# Sustainability of Urban Parks: Applicable Methodological Framework for a Simple Assessment

Teresa González [1], Pia Berger [1,*], Claudia N. Sánchez [1] and Faezeh Mahichi [2]

[1] Facultad de Ingeniería, Universidad Panamericana, Josemaría Escrivá de Balaguer 101, Aguascalientes 20296, Aguascalientes, Mexico; 0230901@ags.up.mx (T.G.); cnsanchez@up.edu.mx (C.N.S.)

[2] Asia Pacific Studies (APS), Ritsumeikan Asia Pacific University (APU), 1-1 Jumonjibaru, Beppu 874-8577, Japan; fmahichi@apu.ac.jp

[*] Correspondence: pberger@up.edu.mx

**Abstract:** Urban parks are central to advancing urban sustainability and improving overall quality of life by providing green spaces that promote physical and mental well-being, mitigate environmental issues, and foster community cohesion. However, there is a lack of methodologies that measure these benefits and provide a sustainability rating. In this study, we propose a valuable tool for measuring the sustainability level of urban parks: low (0–50%), medium (51–79%), and high (80–100%). It employs effective and affordable measures for the daily management of urban parks. It is rooted in the three pillars of sustainability: environmental, social, and economic. We have defined 19 indicators (e.g., renewable energy and energy efficiency, environmental impact on society) and 50 criteria (e.g., clean energy generation, water workshops). A multi-criteria analysis facilitated the selection process for these indicators and criteria. This methodology is developed by characterizing and systematically documenting the park's day-to-day operations. We present a case study of Cárcamos Park in Guanajuato, Mexico. Through this real-life scenario, we demonstrate our methodology's high applicability and effectiveness. The sustainability assessment of Cárcamos Park reveals a level of 57%, with the environmental pillar at 47.7%, the economic pillar at 49%, and the social pillar at 75%. The adaptability of our methodology during the design phase of new parks plays a crucial role in shaping sustainable park layouts. Park managers can apply our procedure to any park, evaluate their sustainability status, and detect areas of opportunity.

**Keywords:** sustainability; urban parks; green areas; sustainable cities

## 1. Introduction

In urban planning and sustainability, the significance of urban parks cannot be overstated. This relevance amplifies as the world is becoming increasingly urbanized [1,2]. Population growth and migration have been the main drivers of the urban shift that has characterized the last century, and this trend is likely to continue in the coming decades [3–5]. The urban share worldwide will rise from around one-third in 1950 to approximately two-thirds in 2050. Sustainable development depends on successfully managing urban growth to create sustainable cities in developed and developing countries [6–8]. The sustainability of cities and their regeneration strategies principally focus on improving the cities' infrastructure and resilience of the urban environment [9,10]. The fundamental goal of a sustainable city is the advancement and facilitation of the long-term well-being of people and the planet through the effective use of natural resources and management of wastes while enhancing liveability through economic prosperity and social well-being within a city [11–13]. A comprehensive definition of sustainable cities and their sustainable development is provided by Roosa [14] (p. 44): "Sustainable development is the ability of physical development and environmental impacts to endure long-term habitation on the planet Earth by human and other indigenous species while providing: (1) an opportunity

for environmentally safe, ecologically appropriate physical development; (2) efficient use of natural resources; (3) a framework which allows improvement of the human condition and equal opportunity for current and future generations, and; (4) manageable urban growth". Sustainable urban planning and the design of green infrastructures such as street trees, green roofs, and open green spaces like parks can contribute to reducing temperature and pollution in urban areas as well as creating habitats to protect biodiversity [15,16].

Urban parks are public places that provide essential ecosystem services, such as oxygen production, air, and water purification [17], as well as noise and air filtering. Parks create a micro-climate and give space for biodiversity protection [18]. In addition, they provide social and psychological services and promote the well-being and education of citizens, which is relevant to the livability of cities [9,19,20].

People often associate urban parks with sustainability in cities. Nevertheless, despite the apparent benefits, the park's presence alone does not automatically imply a positive impact on the environment, society, and economic viability. These spaces must offer specific characteristics that adapt to their location to provide tangible benefits [21]. Sustainable parks are different from traditional parks in three main ways: (1) they are self-sufficient in terms of efficiency in resources like fertilizers, as well as energy and water consumption with regard to reducing maintenance costs; (2) they mitigate greenhouse gas emissions to reduce environmental challenges in cities since they provide sustainable benefits inside their limited boundaries, acting as "green lungs" in their communities; and (3) they provide a habitat for native species [22]. The research of Vélez Restrepo [23] indicates that the contribution of parks, in terms of sustainability and resilience, takes into account the park's energy and water consumption and waste management.

Local sustainable management practices will have global impacts, primarily due to the forces of globalization [24,25]. In addition, local sustainable park practices act as showcases for others and can thus spread. To raise sustainability in urban parks, an increased capacity of responsible managers is essential [26]. Nevertheless, those managers often lack sufficient skills and tools to ensure that local green areas are more resilient to the challenges posed by global change [9].

Furthermore, urban green areas worldwide frequently suffer from economic challenges and are not financially self-sufficient. The economic demands of a park encompass a spectrum of financial obligations that include but are not limited to staff compensation, electricity consumption, maintenance expenses, the establishment and maintenance of green space, the provision of composting facilities, and the construction and maintenance of infrastructure and buildings. Consequently, their maintenance and sustainable development is limited [27,28]. It is therefore essential to improve the management of the parks with practical strategies and tools; continuous collecting and monitoring of information on the condition of parks is fundamental to maintaining environmental quality [29]. The high population density of the cities and the limited recreation areas put high pressure on parks and affect their sustainability. The challenge for urban park managers is to meet the needs of all visitors and still guarantee the sustainability and protection of park resources [30].

According to Dearden et al. [31] and Gavrilidis et al. [32], properly managing and protecting existing parks is more important than creating new parks. In the same way, the tools and strategies in park management are essential to reach and improve upon the goals and targets for present parks [33]. The parks must be designed considering the current and expected future climate and conditions [34]. However, cities need more information on the quantity and quality of urban parks; the existing data need to be completed and more interrelated, and a database with the parks managing information might improve their function [9]. To implement strategies for sustainability, sustainability indicators are essential since they help to measure green areas' functions comprehensively. Sustainability indicators facilitate the assessment of the level of sustainability and the understanding of areas of opportunities for decision-makers and environmental-policy-makers [29,35]. There are some presentations of methodologies to measure the sustainability of urban green areas, none of which can provide a complete picture of sustainability. Social and economic elements are given little

consideration. Cranz and Boland [22] consider five elements to define a park as sustainable: native plants, permeable surfaces, ecological restoration, green infrastructure, and resource self-sufficiency. They defined park sustainability considering social and environmental elements like human and ecological health, environmental education, and wildlife protection [22]. Nevertheless, they neither considered the infrastructure and buildings installed within the park nor information on the park employees or infrastructure, including waste management infrastructure like a waste collection center. Ávila and Medina [36] address the sustainability from a socio-environmental perspective, and Morales-Cerdas et al. [35] include environmental and socio-environmental aspects by applying the following 11 environmental indicators: (1) the percentage of the area in which the protection surface was respected according to the regulations; (2) the percentage of native and exotic species in urban parks; (3) the number of trees per area (density); (4) the species structure, such as the height of trees; (5) the diameter; (6) the number of trees planted in streets; (7) the number of trees planted in street pavements; (8) the number of trees planted in avenues; (9) the soil permeability; (10) the soil biotic index; and (11) the potential of urban parks to host bird-life, for managing urban green areas to determine their environmental condition without considering the economic value. The socio-environmental perspective is critical since it relates the affecting components of visitors to conservation practices in public spaces. In addition to the participation and education of citizens and workers, we propose to include the economic situation (economic pillar) with equal weight as it may reflect efficiency in the use and consumption of resources, leading to economic self-sufficiency. Waste separation and recovering the value of the residues can generate financial resources that can be invested in the park to improve its operation.

Dizdaroglu [13] considers a complete spectrum and notes 10 core sustainable design objectives of urban parks, which are (1) providing green infrastructure; (2) creating a place for people of all ages; (3) building connected park systems within walking distance; (4) implementing water and energy conservation practices; (5) waste management; (6) promoting access to fresh, healthy, and low-cost food; (7) supporting and preserving biodiversity; (8) environmental education and stewardship through hands-on activities; (9) ensuring the long-term maintenance and management of the park; and (10) supporting disaster resilience. Within these objectives, they describe, in theoretical terms, the importance of sustainable park design and management as it broadens the scope of parks in the role of sustainable cities in helping to overcome environmental problems arising from urban sprawl. In order to measure the degree of sustainability of universities and identify their areas of opportunity, one might use the UI Green Metric World University Ranking (Green Metrics) [37]. The Green Metrics initiative, launched by the University of Indonesia in 2010, provides the result of an online survey regarding the current conditions and policies related to green campuses and sustainability in universities worldwide. Green Metrics has six criteria: setting and infrastructure, energy and climate change, waste, water, transportation, education, and research, with 51 indicators that focus on the objectives of sustainable universities. It is a simple guide to measuring and applying university sustainability. Nonetheless, nothing similar to the Green Metric World University Ranking exists for parks [38].

Our methodology focuses on integrating all components related to the operation of an urban park, which is essential for a complete survey. To this end, we have developed a scalable, flexible, and replicable tool that enables the measurement of the sustainability of urban parks. Our methodology is designed to be reproducible, low-cost, and easy to implement by anyone using collected operational data from the park. Our objective is to obtain a sustainability grade for the park and identify areas of improvement. To achieve this, we have developed a method based on three dimensions of sustainability, which we call pillars: environmental, social, and economic. These three pillars contain 50 criteria and 19 indicators that are used to characterize and gather information from day-to-day operations. Using our methodology, park managers can plan short-, medium-, and long-term environmental, social, and financial actions while tracking their progress.

Furthermore, our indicators can be used as a reference for designing new urban parks that are sustainable from the outset.

## 2. Methodology and Sustainability Scheme Proposal

Our methodology corresponds to a quantitative analysis of park operation data, which seeks to draw a scalable, flexible, and replicable roadmap in other parks. This procedure consists of creating a database on the park's operation, including environmental, economic, and social aspects. Konijnendijk et al. [20] define urban parks as "delineated open space areas, mostly dominated by vegetation and water, and generally reserved for public use. Urban parks are mostly larger but can also have the shape of smaller 'pocket parks'. Urban parks are usually locally defined (by authorities) as "parks" ([20], p. 2). Our methodology exhibits broad applicability across urban parks of varying sizes and diverse characteristics. However, it is important to note that there is little evidence to suggest that our methodology can be effectively adapted to small parks, lawns, sports parks, waterside parks, and similar settings. Therefore, our methodology is primarily limited to larger urban parks characterized by a diverse range of facilities and infrastructure components, including buildings and recreational areas.

**Pillars, indicators, and criteria:** Our methodology is based on the three pillars of sustainability: environmental, economic, and social. For facilitating the data collection and analysis, the three pillars are divided into 19 indicators that consist of 50 criteria (the database) (see Figure 1).

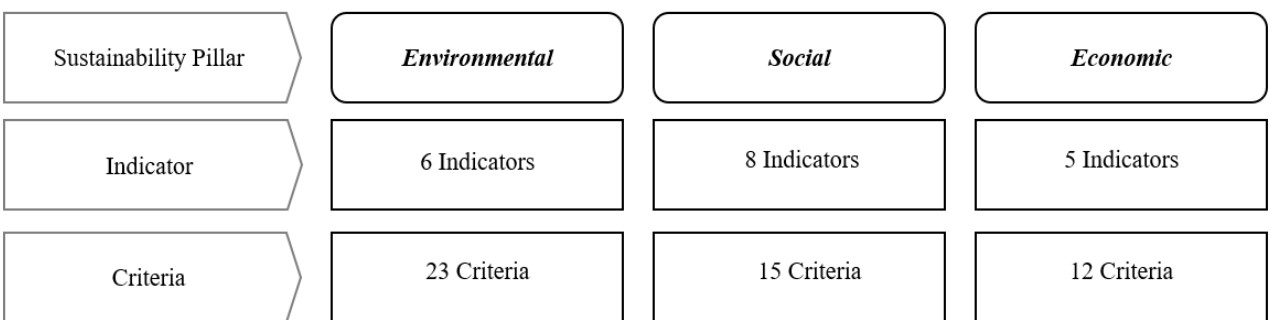

**Figure 1.** Sustainability scheme.

The inclusion of the specific criteria and indicators in the data collection for measuring sustainability is essential due to the following rationales: resource use (quantify and conserve resources), financial health (ensure economic viability), environmental impact (reduce ecological footprint), social well-being (enhance community and visitor experiences), biodiversity (preserve ecological integrity), resilience (adapt to environmental challenges), innovation (drive continuous improvement), equity (promote accessibility and fairness), education (engage and educate the public), and compliance (adhere to legal standards) [9,12–14]. These points, which offer a holistic view of sustainability in the park, are important for measuring sustainability and are therefore taken into account in our indicators and criteria described below.

Each pillar is evaluated equally, with one-third each, no matter the number of indicators or criteria. The highest sustainable value that an urban park can achieve is 100%.

### 2.1. Indicators and Criteria

The indicators represent a first differentiation of the pillars that represent specific groups or topics of the criteria.

Criteria of the same category are joined to one indicator of 19 indicators in total. The environmental pillar consists of 6 indicators and 23 criteria. The indicators are: (E1) sustainable transport, (E2) green area and biodiversity, (E3) water conservation, (E4) renewable energy and energy efficiency, (E5) waste management, and (E6) sustainable building with

certification (sustainability) (see Figure 1). The social pillar consists of 8 indicators and 15 criteria. The indicators are: ((S1) exclusive maintenance staff, (S2) environmental impact on society, (S3) space for environmental education, (S4) environmental policies for the use of the green area, (S5) environmental policies for the use of the green area, (S6) environmental management system in the office, (S7) accessible entrance, and (S8) sustainable building with certification (health) (see Figure 1). Finally, the economic pillar has 5 indicators and 12 criteria. The indicators are: (EC1) sale of waste, (EC2) charging fees, (EC3) waste registration and collector control, (EC4) energy efficiency, and (EC5) sustainable building with certification (efficiency) (see Figure 1). The social pillar has the highest number of indicators (8). The environmental pillar has the highest number of criteria (23). The economic pillar has the lowest number of criteria (12) and indicators (5).

### 2.2. Criteria and Additional Data

The criteria represent numerical information collected by the park management on a day-to-day basis that directly influences the grade of sustainability. Some of these criteria (collected data) need additional data to be calculated (see Table 1). Those additional data do *not* have a direct impact on the level of sustainability. For example, the criterion (1) *percentage of park employees that use car mobility to go to work* is calculated from two additional data: (1) the additional data *number of park employees that use car mobility to go to work* and (2) the additional data *total number of persons* (see Figure 1 and Table 1). The 18 additional data are only required to evaluate the environmental and economic pillar criteria. No additional data are required for the social pillar criteria (see Table 1).

**Table 1.** Additional data used for assessing certain criteria (1, 2, 3, 4, and 5) in the indicators of the environmental pillar (E1, E2, E3, and E4) and economic pillar (EC4).

| No. | Additional Data | Indicator/Criteria |
|---|---|---|
| 1 | Total number of employees | E1/1 |
| 2 | Number of employees that use car mobility to go to work | E1/1 |
| 3 | Total surface of the park ($m^2$) | E2/1, E2/3 |
| 4 | Total surface of green area ($m^2$) | E2/1 |
| 5 | Total surface of constructed and sealed area ($m^2$) | E2/1 |
| 6 | Total number of trees in the green area | E2/3, E2/4, E2/5 |
| 7 | Number of native trees | E2/4 |
| 8 | Number of healthy trees | E2/5 |
| 9 | Volume of treated water ($m^3$) | E3/3 |
| 10 | Volume of water used for irrigation ($m^3$) | E3/3 |
| 11 | Number of total bulbs in the offices | E4/1, EC4/2, EC4/3 |
| 12 | Number of LED bulbs in the offices | E4/1, EC4/2, EC4/3 |
| 13 | Number of total bulbs in the green area | E4/2 |
| 14 | Number of LED bulbs in the green area | E4/2 |
| 15 | Total energy consumption (kWh) | E4/3, E4/4, EC4/1 |
| 16 | Energy produced by renewable energies | E4/3, EC4/4, EC4/1 |
| 17 | Total number of electronic equipment | EC4/4 |
| 18 | Number of electronic equipment that are less than five years old | EC4/4 |

**Positive or negative impact:** Criteria were determined to have a positive or negative impact on sustainability. Positive impacts were highlighted with a plus sign (+), and negative impacts have been assigned a negative sign (−).

**Internal and external impacts (dependencies):** Each criterion was classified according to its internal or external dependency to reveal if the park management has direct control of the value of the criterion (internal) or if the criterion cannot be directly influenced by the park management (external). The impacts or dependencies of internal control are criteria the park manager can intervene or modify, such as automatic irrigation. Those internal dependencies are marked with the letter (i+) if the impact is positive and (i−) if the impact is negative.

The criteria of external dependencies are criteria where only external forces like the government can intervene or modify, not the park management itself. For example, the criterion "presence of a lake (water body) inside the park" is marked with the letter (e+) since the decision to install a lake is external and a lake has a positive impact on sustainability. The criterion "number of park employees that use car mobility to go to work" was assigned with the impact "i−" since this criteria can be influenced internally by the park management (i, Internal) and has a negative (−) impact on sustainability.

**Sustainability degree:** The first step in achieving the sustainability degree is assigning the value for each criterion. Those values range from 0 to 100 points, with 100 being the highest possible. For a positive impact criterion (+), 100 is the best sustainability value. In the case of a negative impact criterion (−), 0 is the best value for a high sustainability assessment.

The indicator weights result from the average of the corresponding criteria values. Then, the weights of the pillars are computed as the average of their indicator values. The final sustainability degree is calculated by averaging the values of all three pillars, with each pillar participating equally with one-third.

According to the appraisal and evaluation strategies proposed by Dodgson et al. [39] (Chapter 2, pages 9–13), the decision-making process that helped us to define the point range of the degrees of sustainability (see Table 2) was as follows: (1) identifying objectives, (2) identifying options for achieving the objectives, (3) identifying the criteria to be used to compare the options, (4) analysis of the options, (5) making choices, and (6) feedback [39].

The three sustainability levels and their point range are defined as: (1) low sustainability for those with a total score between 0 and 50, (2) medium sustainability with a score between 51 and 79, and (3) high sustainability for a score between 80 and 100 (see Table 2).

**Table 2.** Degrees of sustainability in parks.

| Point Range | Sustainability |
|---|---|
| 0–50 | Low sustainability |
| 51–79 | Medium sustainability |
| 80–100 | High sustainability |

### 2.2.1. Criteria in the Environmental Pillar

The environmental pillar contains 6 indicators and 23 criteria (see Table 3).

**Table 3.** Sustainability values: environmental pillar with its respective *impact* (internal i or external e, positive + or negative −), *current park value* (recent sustainability value of Cárcamos Park, Mexico), and *parks improvement value* (percentage of potential improvement considering only internal impacts that can be influenced by park managers).

| Indicator (E1–E6), Criteria (1, 2, . . . , 6) | Impact Internal (i), External (e), Positive (+), Negative (−) | Current Park Value | Parks Improvement Value |
|---|---|---|---|
| **E1: Sustainable transport** | | **8.8** | **50.0** |
| 1. Percentage of park employees that use car mobility to go to work | i− | 35.0 | 100.0 |
| 2. Kilometers driven per day per employee to go to the park | e− | 0.0 | 0.0 |
| 3. Low-emission motorised transport | e+ | 0.0 | 0.0 |
| 4. Bicycle infrastructure | i+ | 0.0 | 100.0 |

Table 3. *Cont.*

| Indicator (E1–E6), Criteria (1, 2, . . . , 6) | Impact Internal (i), External (e), Positive (+), Negative (−) | Current Park Value | Parks Improvement Value |
|---|---|---|---|
| **E2: Green area and biodiversity** | | **64.4** | **92** |
| 1. Percentage of green area | e+ | 60.0 | 60.0 |
| 2. Pollinator garden | i+ | 100.0 | 100.0 |
| 3. Trees per hectare | i+ | 20.2 | 100.0 |
| 4. Percentage of native trees | i+ | 54.6 | 100.0 |
| 5. Percentage of healthy trees | i+ | 87.3 | 100.0 |
| **E3: Water conservation** | | **33.3** | **66.6** |
| 1. Presence of a lake (water body) inside the park | e+ | 100.0 | 100.0 |
| 2. Automatic irrigation | i+ | 100.0 | 100.0 |
| 3. Use of treated water for irrigation | e+ | 0.0 | 0.0 |
| 4. Percentage of water treated after use | e+ | 0.0 | 0.0 |
| 5. Rainwater harvesting systems | i+ | 0.0 | 100.0 |
| 6. Water-saving devices | i+ | 0.0 | 100.0 |
| **E4: Renewable energy and energy efficiency** | | **96.5** | **100.0** |
| 1. Percentage of LED lighting in the offices | i+ | 92.0 | 100.0 |
| 2. Percentage of LED lighting in the green area | i+ | 100.0 | 100.0 |
| 3. Clean energy generation | i+ | 94.0 | 100.0 |
| 4. Emission reductions from clean energy generation | i+ | 100.0 | 100.0 |
| **E5: Waste management** | | **83.3** | **100.0** |
| 1. Waste collection and separation service | i+ | 100.0 | 100.0 |
| 2. Organic waste management (composting) | i+ | 100.0 | 100.0 |
| 3. Recycling program | i+ | 50.0 | 100.0 |
| **E6: Sustainable building with certification (sustainability)** | | **0.0** | **0.0** |
| 1. Sustainable building with green building certification | e+ | 0.0 | 0.0 |
| **Total value of environmental pillar** | | **47.7** | **68.1** |

*Sustainable transport indicator* (E1) considers four different criteria, and the value of this indicator consists of their average. Criterion (1) *percentage of park employees that use car mobility to go to work* refers to the percentage of workers who come to the park with their car that runs on fossil fuels. This percentage results from the two additional data: *total number of employees* and *number of employees that use car mobility to go to work*. This resulting percentage equals the number of points for this criterion. Criterion (2) *kilometers driven per day per employee to go to the park* refers to the distance between work and home that park employees must travel and includes only fossil fuel cars. The daily trips should be optimized and be a maximum of 7 km from home to work to accelerate the urban development solution and sustainable mobility [40]. Criterion (3) *low-emission motorized transport* represents the possibility of reaching the park with a low-emission means of transport (e.g., public transport or organizing car sharing). The existence of a low-emission transport to the park leads to 100 points. The contrary leads to zero points. Criterion (4) *bicycle infrastructure* refers to the availability of bicycles in the park to encourage their use as an alternative means of transportation. If bicycles are available inside the park, 100 points; if not, 0 points. *Green area and biodiversity indicator* (E2) considers the values of five criteria whose average leads to the indicator value: Criterion (1) *percentage of green area* is determined using three additional data: the *total surface of the park*, the *total surface of green area*, and *total surface of constructed and sealed area* (see Table 1). The number of points available in this indicator is equal to the percentage of the park's green area. Criterion (2) *pollinator garden* evaluates if there is a pollinator garden inside the park. If the park counts

with a pollinator garden, this equals 100 points. No pollinator garden equals 0 points. Criterion (3) *number of trees per hectare* takes into account a minimum number of trees per hectare in parks. Therefore, we adhered to the guidance stipulated by the National Forestry Agency in Mexico (CONAFOR), which recommends a tree density of 625 trees per hectare for parks [41], which refers to one tree every four meters. This criterion could be adapted to the recommended data from organizations outside Mexico. The points that can be achieved for this criterion are a percentage of this recommended number of trees per hectare. The value of criterion (4) *percentage of native trees* results from the percentage of native trees (calculated from two additional data: *total number of trees* and the *number of native trees*, see Table 1) from the recommended of 80% of native trees (and maximum 20% exotic trees), according to Sánchez and Artavia [42]. Criterion (5) *percentage of healthy trees* is calculated including the additional data of the *total number of trees* and *number of healthy trees* (see Table 1. The percentage of healthy trees in the park equals the score of this criterion. The average of all criteria leads to the value of this indicator.

*Water conservation indicator* (E3) considers six criteria. Criterion (1) *presence of a lake (water body) inside the park* scores 100 points if the park has a lake; no lake scores zero points. Criterion (2) *automatic irrigation* scores 100 points if the park uses nutrient-rich water from the lake to irrigate green areas. Criterion (3) *use of treated water for irrigation* leads to 100 points if the park uses treated water for irrigation, regardless of where the water was treated, outside or inside the park. Criterion (4) *percentage of water treated after use* refers to the percentage of irrigation water that was treated after being used in bathrooms or other facilities, etc. This percentage equals the number of points. Two additional data are necessary for this criterion: *volume of treated water* and *volume of water used for irrigation*. Criterion (5) *rainwater harvesting systems* means that installing a water-capturing facility leads to 100 points; if the park lacks such a technology, zero points are awarded. Criterion (6) *water-saving devices* considers installing water-saving technologies in bathrooms: its percentage equals the number of points.

*Renewable energy and energy-efficiency indicator* (E4) considers four criteria. Additional data (11–16) are necessary to appraise the criteria of this indicator (see Table 1). Their average equals the value of this indicator: criterion (1) *percentage of LED lighting in the offices* refers to the percentage of LED lamps installed in the buildings equals the number of points. The two criteria that lead to this value are the total number of bulbs and the number of LED bulbs installed in the buildings. Criterion (2) *percentage of LED lighting in the green area* refers to the percentage of LED lamps installed in the green area equals the number of points (the total number of bulbs and the number of LED bulbs installed in the green areas are the two criteria that lead to this value). Criterion (3) *clean energy generation* refers to the generation of green energy by renewable energies like solar or wind. Criterion (4) *emission reductions from clean energy generation*; their present percentage leads to the number of points.

*Waste management indicator* (E5) evaluates the management of residues inside the park and considers three criteria. The criterion (1) *waste collection and separation service* indicates if the park offers a recycling center where recyclable waste like paper, metal, glass, batteries, or PET (polyethylene terephthalate) are collected and separated. The presence of such a service leads to 100 points; the contrary equals zero points. Criterion (2) *organic waste management (composting)* leads to 100 points if the organic waste from the park is collected and composted. Criterion (3) *recycling program* leads to 100 points if a recycling program helps prevent waste and regulates its treatment in a sustainable way. No recycling program in the park would lead to zero points.

*Sustainable building with certification (sustainability) indicator* (E6) includes one criterion, criterion (1) *sustainable building with green building certification*. If the park has a building, it must have a green building certificate, such as LEED, Passive House, or BREEAM [43,44]. In order to obtain 100 points in this criteria, a green building certificate guarantees the sustainability of the building [45]. No certificate leads to 0 points. If the park has no building, this indicator counts for 100 points. If the park has more than one building, the percentage of buildings with green certificates equals the number of points.

### 2.2.2. Criteria in the Social Pillar

The social pillar holds 8 indicators and 15 criteria.

*exclusive maintenance staff indicator* (S1) takes into account one criterion: criterion (1) *Exclusive maintenance staff*, which refers to employees that take care of and maintain the sustainable aspects of the park. At least one employee who takes care of the sustainable aspects inside the park, reflected by the three pillars (environmental, social, and economic), leads to 100 points.

*Environmental impact on society indicator* (S2) considers one criterion: criterion (1) *environmental education events* refers to environmental education events offered by the park, which should be at least 12 events per year or one per month. That means 12 events per year equals 100 points; 6 events leads to 50 points; and no events leads to zero points.

*Space for environmental education indicator* (S3) considers one criterion: criterion (1) *space to promote environmental education* refers to the existence of a dedicated space for environmental education activity. In order to achieve the highest score, the park has such an area available, which can be, for example, a botanical garden or a butterfly house.

*Environmental education workshops indicator* (S4) takes into account six criteria, which are public information lectures or workshops on the most important environmental issues for everyone: criterion (1) *biodiversity workshops* includes talks and/or activities on biodiversity, criterion (2) *waste workshops* includes talks/activities on resources and residues, criterion (3) *air quality workshops* includes talks and/or activities on air quality, criterion (4) *soil workshops* includes talks/workshops on the importance of the soil, criterion (5) *water workshops* represents talks and/or activities on water protection and criterion (6) *climate workshops* represents workshops on the importance of climate change. One talk and/or activity on a respective topic equals 100 points. No activity equals zero points. The average of all criteria leads to the value of the indicator.

*Environmental policy for the use of green area indicator* (S5) considers criterion (1) *environmental policies for use of green area*. If the park counts with environmental policies like the visitors' behavior in an environmentally friendly way (avoiding single-use containers or giving instructions on how to take care of flora and fauna inside the park), it leads to 100 points. No established environmental policy leads to zero points.

The *environmental management system in office indicator* (S6) consists of criterion (1) *environmental management system in office*. If the park has implemented a program, this criterion obtains the maximum score of 100 points.

*Accessible entrance indicator* (S7) considers three criteria defining the accessibility of the park to all citizens. Criterion (1) *free access (no entrance fee)* means that access to the park is free and no entrance fee is charged. Criterion (2) *open 7 days a week* means that the park opens every day (365 days/year), and criterion (3) *open at least 10 h a day* considers that the park is available for the public for a minimum of 10 h per day. In conclusion, all three criteria would lead to 100 points; if the criteria are not fulfilled, this equals to zero points for the respective criteria. Their average leads to the value of this indicator.

*Sustainable building with certification (health) indicator* (S8) includes criterion (1) *sustainable building for healthy living/working*. If the park has a building, it must have a green building certificate, such as LEED, Passive House or BREEAM [43,44], in order to obtain 100 points in this criterion since a green building certificate guarantees a healthy atmosphere inside the building [45]. No certificate leads to 0 points. If the park has no building, this indicator counts for 100 points. If the park has more than one building, the percentage of buildings with a green certificate equals the number of points.

### 2.2.3. Criteria in the Economic Pillar

The economic pillar contains 5 indicators and 12 criteria that consider economic resources.

Economic resources generated from *sale of waste indicator* (EC1) considers two criteria of economic value: (1) *sale of paper, carton, plastic, aluminum, iron, newspaper, electronic, tetra-pack, organic waste, and glass*. Plastics refers to PET since it is economically the most important [46,47]. For each residue in the list, the park receives 10 points. If all waste types

are sold, 100 points are achieved. Criterion (2) *alkaline batteries* shows the importance of collecting alkaline batteries and guaranteeing their adequate recycling. Both criteria are of external impact since the park management cannot influence the price of residues. The average of all criteria values leads to this indicator's value.

*Charging fees indicator* EC2 considers three criteria. Criterion (1), *entrance fee*, indicates if the entrance to the park is free of charge. Criterion (2) *workshop fee* indicates if the park can generate economic revenue by organizing and charging for workshops. Criterion (3) *rent space fee* indicates if the park can generate economic revenue by charging for the rent of special areas inside the park. All three criteria lead to zero points if no fees are charged since this would be an economic disadvantage.

*Waste registration and collector control indicator* (EC3) considers two criteria: criterion (1) *waste registration* of collected and separated waste at the collection center, including the type of waste, the weight (kg), and the distance (m) from where they come from; the presence of a waste registration folder (analog or digital) leads to 100 points, and no continuous waste registration leads to zero points; and criterion (2) *authorized waste collector*; the park obtains 100 points if the waste collector is legally authorized and counts with all necessary permits to manage recyclable waste. The average of both criteria leads to the indicator's value.

*Energy-efficiency indicator* (EC4) considers four criteria: criterion (1) *clean energy generation* refers to economic savings through renewable energy generation like solar, wind, geothermic, and biomass plants. The percentage of green energy generation leads to the number of points. Criterion (2) *percentage of LED-illumination in offices* refers to the percentage of highly efficient illumination (LED) in the offices that leads to the number of points. Criterion (3) *percentage of LED-illumination in the green area*, refers to the percentage of highly efficient illumination (LED) in the green area leading to the corresponding number of points. Criterion (4) *efficient electronic equipment (not older than 5 years)* means that recent energy-efficient devices and equipment like pumps and computers, printers, and refrigerators that are less than five years old can lead to economic benefits; the percentage of the respective installations leads to the number of points. All criteria of this indicator need additional data (see Table 1 to determine the criteria of this indicator. The indicator's value represents the average of all criteria values.

*Sustainable building with certification (efficiency) indicator* (EC5) includes one criterion: (1) *sustainable building for energy efficiency and cost-savings*. If the park has a building, it must have a green building certificate, such as LEED, Passive House, or BREEAM [43,44], in order to obtain 100 points in this criterion since a certificate guarantees cost savings through energy savings and cost efficiency [48]. No certificate leads to 0 points. If the park has no building, this indicator counts for 100 points. If the park has more than one building, the percentage of buildings with a green certificate equals the number of points. In the following section, we demonstrate the applicability of our methodology and its ease of assessment using real operational data from Cárcamos Park.

## 3. Results

This Section presents our methodology's application in Cárcamos Park in the city of Leon in Guanajuato, Mexico. Cárcamos Park was selected as the study site due to unrestricted access to the entirety of the park's operational dataset. First, we describe the park we used as a case study and then explain the captured operational data. Finally, we show the sustainability values obtained for each criterion, indicator, and pillar and calculate the sustainable degree of this case study park. The study's outcome identified the opportunities for improving the sustainability level of Cárcamos Park.

### 3.1. Case Study Cárcamos Park, Mexico

In order to test the applicability of our methodology, we introduced the real operational and maintenance data from Cárcamos Park to our datasheet. Cárcamos Park is located in the City of León in Guanajuato, Mexico (see Figure 2). Cárcamos Park has a total area

of 116,074.99 m$^2$, of which the green area occupies 60% with 1457 trees; the built-up area occupies 4%, and a lake occupies 36% of the surface area.

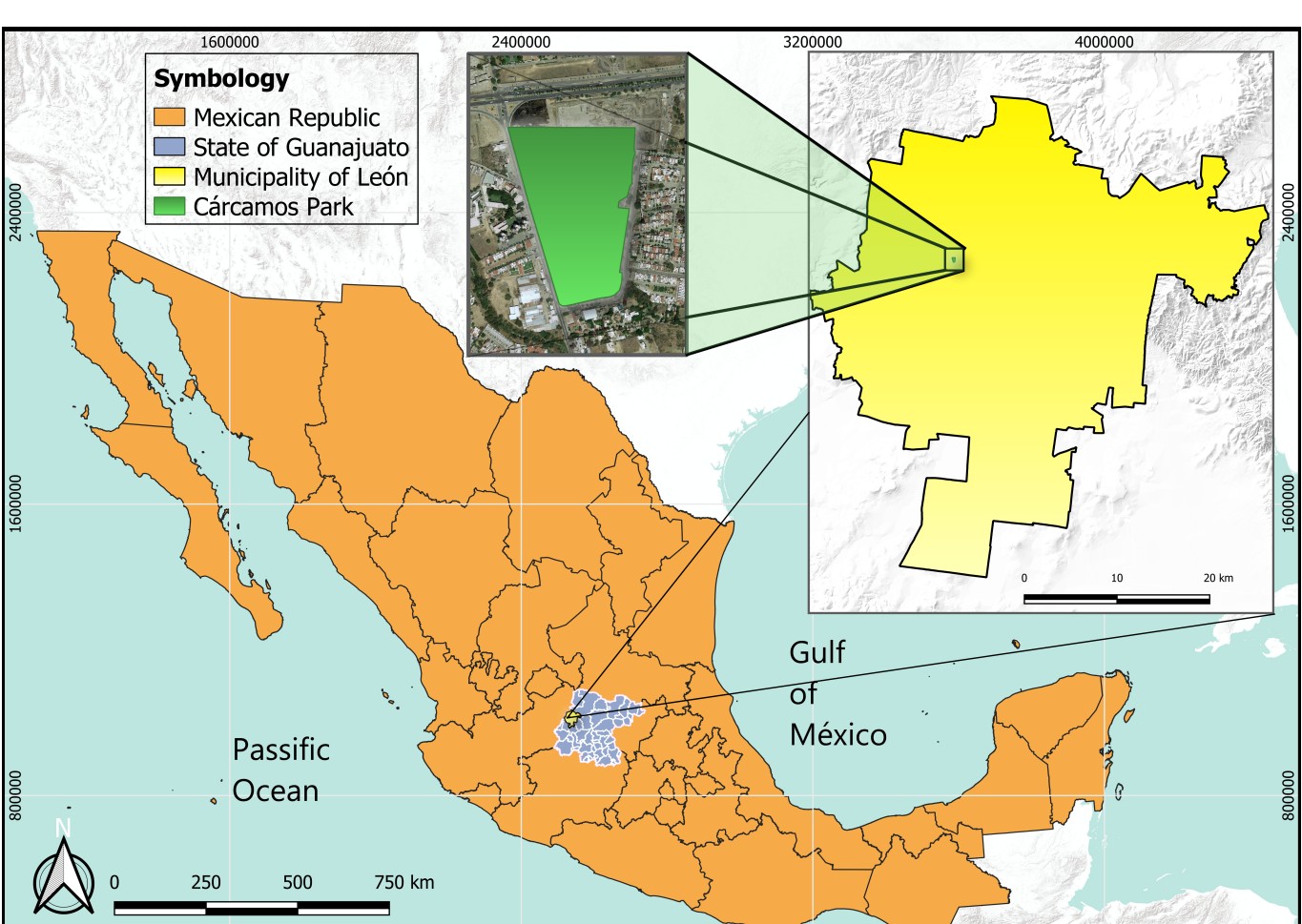

**Figure 2.** Location of Cárcamos Park in Mexico. Source: Own elaboration using QGis software and INEGI layers [49].

Cárcamos Park serves a dual purpose. It provides ample public green space and accommodates buildings with government offices in the park's southern area. The presence of government employees in the park has numerous benefits, particularly in data collection and monitoring.

The government building inside the park spans two floors, which shelters a team of 20 employees who work weekdays from Monday to Friday. In addition to this, the park features an area dedicated to promoting environmental education and a collection center for citizens to drop off waste materials with a monetary value. These facilities are open every day of the year from 6:00 a.m. to 8:00 p.m., offering park visitors 14 h of access per day. The park also employs a full-time maintenance worker to tend to the green space, while the government office staff stationed on-site split their duties between the botanical garden and the collection center.

We have carefully compiled all relevant operation data of Cárcamos Park and summarized it in a comprehensive data sheet with 50 criteria and 18 additional data (see Table 1). In order to ensure a complete understanding of the park's performance, we found it necessary to collect information over one year spanning all seasons, including changing seasons and situations such as holiday seasons, rainy seasons, and droughts. Thus, we incorporated all operational data from January to December 2020 to generate the database. Although 2020 was atypical, marked by the global SARS-COVID-19 pandemic, it provided an opportunity to evaluate the park's behavior from a baseline level. Additionally, with the reactivation of

activities in August, the database facilitated an observation of the movements and changes in the park's operating parameters.

### 3.2. Sustainability Degree of the Case Study, Cárcamos Park, Mexico

This proposal presents the determination of the degree of sustainability assigned for Cárcamos Park in Mexico. It identifies the percentage of potential opportunities for enhancement for park sustainability (see Figures 3–5).

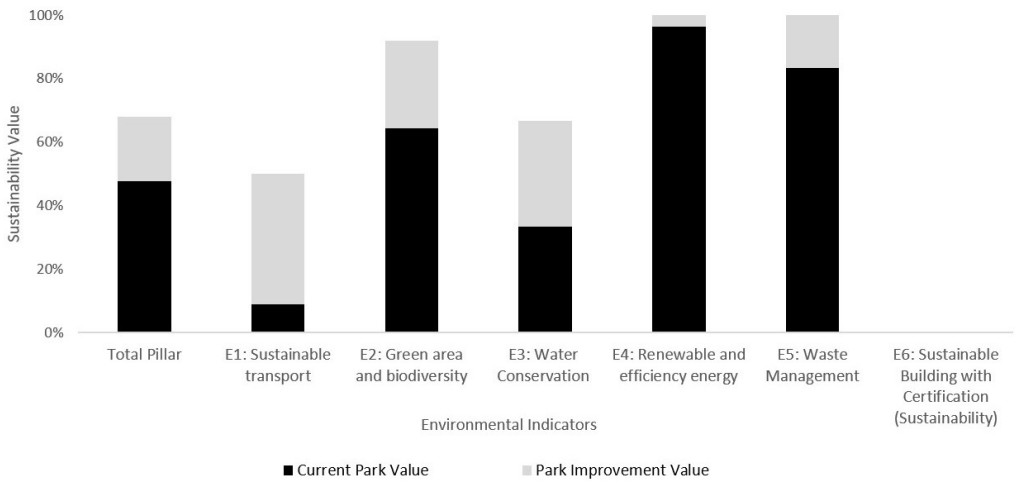

**Figure 3.** Environmental pillar. Sustainability value obtained per indicator for Cárcamos Park, Mexico. *Current park value* (recent sustainability value of Cárcamos Park, Mexico); *parks improvement value* (percentage of potential improvement considering only internal impacts that can be influenced by park managers).

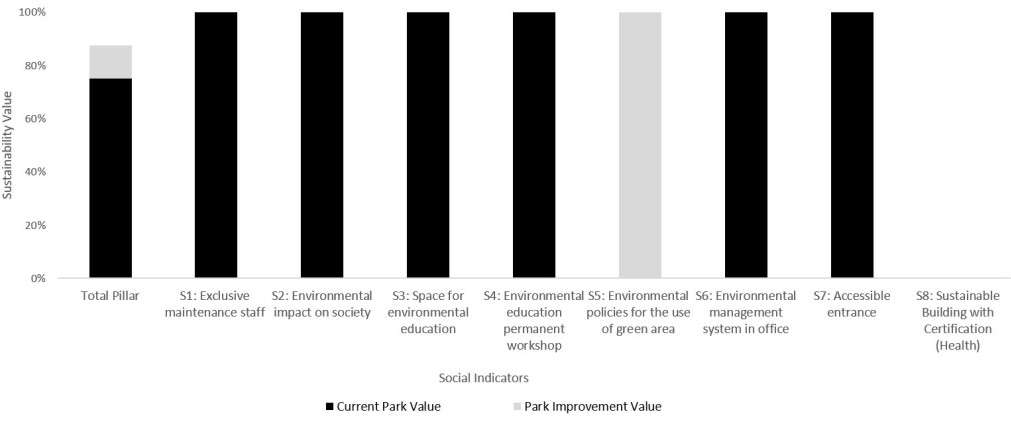

**Figure 4.** Social pillar. Sustainability value obtained per indicator for Cárcamos Park, Mexico. *Current park value* (recent sustainability value of Cárcamos Park, Mexico); *parks improvement value* (percentage of potential improvement considering only internal impacts that can be influenced by park managers).

Once the database with information for each of the 50 criteria and 18 additional data of Cárcamos Park was completed, we determined the values for the criteria. The weights for the indicators and pillars of Cárcamos Park resulted from summing the criteria´s values and taking their average (see Tables 3–6 and Figures 3–5).

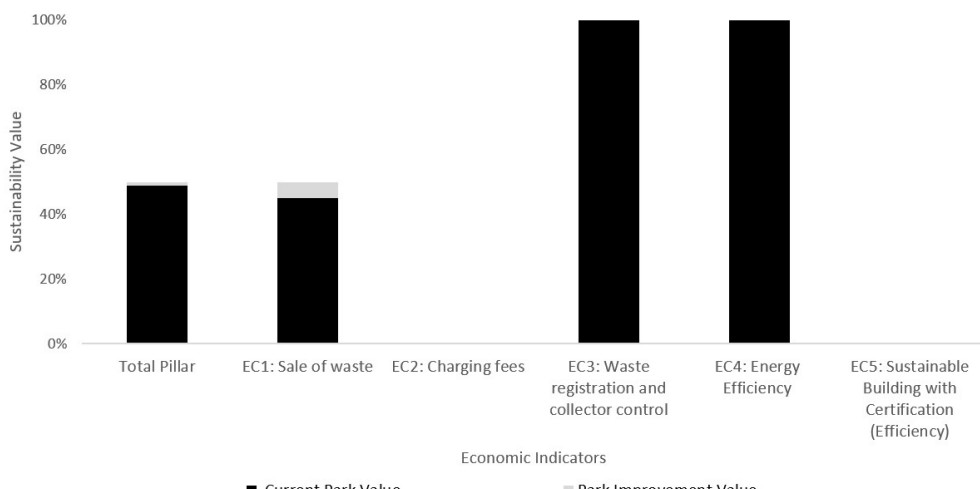

**Figure 5.** Economical pillar. Sustainability value obtained per indicator for Cárcamos Park, Mexico. *Current park value* (recent sustainability value of Cárcamos Park, Mexico); *parks improvement value* (percentage of potential improvement considering only internal impacts that can be influenced by park managers).

**Table 4.** Sustainability Values: social pillar with its respective *impact* (internal i or external e)(positive impact + or negative impact −), *current park value* (recent sustainability value of Cárcamos Park, Mexico), and *parks improvement value* (percentage of potential improvement considering only internal impacts that can be influenced by park managers).

| Indicator (S1–S8), (1, C.; 2; ; 6) | Impact Internal (i), External (e), Positive (+), Negative (−) | Current Park Value | Parks Improvement Value |
|---|---|---|---|
| **S1: Exclusive maintenance staff** | | **100.0** | **100.0** |
| 1. Exclusive maintenance staff | e+ | 100.0 | 100.0 |
| **S2: Environmental impact on society** | | **100.0** | **100.0** |
| 1. Environmental education events | i+ | 100.0 | 100.0 |
| **S3: Space for environmental education** | | **100.0** | **100.0** |
| 1. Space to promote environmental education | i+ | 100.0 | 100.0 |
| **S4: Environmental education workshop** | | **100.0** | **100.0** |
| 1. Biodiversity workshops | i+ | 100.0 | 100.0 |
| 2. Waste workshops | i+ | 100.0 | 100.0 |
| 3. Air quality workshops | i+ | 100.0 | 100.0 |
| 4. Soil workshops | i+ | 100.0 | 100.0 |
| 5. Water workshops | i+ | 100.0 | 100.0 |
| 6. Climate change workshops | i+ | 100.0 | 100.0 |
| **S5: Environmental policies for the use of green area** | | **0.0** | **100.0** |
| 1. Environmental policies for use of green area | i+ | 0.0 | 100.0 |
| **S6: Environmental management system in office** | | **100.0** | **100.0** |
| 1. Environmental management system in office | i+ | 100.0 | 100.0 |

<center>**Table 4.** *Cont.*</center>

| Indicator (S1–S8), (1, C.; 2; ; 6) | Impact Internal (i), External (e), Positive (+), Negative (−) | Current Park Value | Parks Improvement Value |
|---|---|---|---|
| **S7: Accessible entrance** | | **100.0** | **100.0** |
| 1. Free access (no entrance fee) | e+ | 100.0 | 100.0 |
| 2. Open 7 days a week | e+ | 100.0 | 100.0 |
| 3. Open at least 10 h per day | e+ | 100.0 | 100.0 |
| **S8: Sustainable building with certification (health)** | | **0.0** | **0.0** |
| 1 Sustainable building for healthy living/working | e+ | 0.0 | 0.0 |
| **Total value of social pillar** | | **75** | **87.5** |

**Table 5.** Sustainability values: economic pillar with its respective *impact* (internal i or external e) (positive impact + or negative impact −), *current park value* (recent sustainability value of Cárcamos Park, Mexico), and *parks improvement value* (percentage of potential improvement considering only internal impacts that can be influenced by park managers).

| Indicator (EC1-EC5), Criteria (1, . . . , 4) | Impact Internal (i), External (e), Positive (+), Negative (−) | Current Park Value | Parks Improvement Value |
|---|---|---|---|
| **EC1: Sale of waste** | | **45** | **50** |
| 1. Sale of paper, carton, plastic, aluminium, iron, newspaper, electronic, tetra-pack, organic waste, and glass | i+ | 90.0 | 100.0 |
| 2. Alkaline batteries | i+ | 0.0 | 100.0 |
| **EC2: Charging fees** | | **0.0** | **0.0** |
| 1. Entrance fee | e+ | 0.0 | 0.0 |
| 2. Workshop fee | e+ | 0.0 | 0.0 |
| 3. Rent space fee | e+ | 0.0 | 0.0 |
| **EC3: Waste registration and collector control** | | **100.0** | **100.0** |
| 1. Waste registration | i+ | 100.0 | 100.0 |
| 2. Authorised waste collector | i+ | 100.0 | 100.0 |
| **EC4: Energy efficiency** | | **98.0** | **100.0** |
| 1. Clean energy generation | i+ | 100.0 | 100.0 |
| 2. Percentage of LED-illumination in offices | i+ | 92.0 | 100.0 |
| 3. Percentage of LED-illumination in green area | i+ | 100.0 | 100.0 |
| 4. Efficient electronic equipment (not older than 5 years) | i+ | 100.0 | 100.0 |
| **EC5: Sustainable building with certification (efficiency)** | | **0.0** | **0.0** |
| 1. Sustainable building for energy efficiency and cost-savings | e+ | 0.0 | 0.0 |
| **Total value of economic pillar** | | **49** | **50** |

Tables 3–6 show the sustainability values of the individual pillars (environmental, social, and economic) with their respective impact (internal i or external e, positive + or negative −), their current park value (recent sustainability value of Cárcamos Park, Mexico), and their parks improvement value (percentage of potential improvement considering only internal impacts that can be influenced by park managers).

**Table 6.** Sustainability value obtained per pillar for Cárcamos Park: *current park value* and *parks improvement value* (internal criteria that can be changed by the park management).

| Pillar | Current Park Value | Parks Improvement Value |
|---|---|---|
| environmental pillar | 47.7 | 68.1 |
| social pillar | 75 | 87.5 |
| economic pillar | 49 | 50 |
| **Total value** | **57.2** | **68.5** |

The Figures 3–5 demonstrate the sustainability values of the individual pillars, which have been assessed per indicator for Cárcamos Park, Mexico; the current park value represents the recent sustainability value of Cárcamos Park, and the parks improvement value refers to the percentage of the potential improvement considering only the internal impacts (i). These points can be influenced directly by the park managers. The indicators (E6) (sustainability) (Figure 3), (S8) (health) (Figure 4), (EC2) charging fees, and (EC5) (efficiency) (Figure 5) do not show any sustainability value nor have a park improvement value since the criteria of these indicators (E6, S8, EC2, and EC5) cannot be influenced by the park managers of Cárcamos Park.

Figure 6 shows the total sustainability values of all three pillars and the three individual sustainability values obtained per pillar for Cárcamos Park, México. The pillar of the highest sustainability value is the social pillar, with 75%, and the pillar of the lowest is the environmental pillar, with 47.7%. The sustainability values of both pillars, the environmental pillar (47.7%) and the economic pillar (49%), are in the range of low sustainability (see Table 2). The average of all three pillars increased the total value of sustainability. Cárcamos Park resulted in a final degree of sustainability of 57.2%, which is classified as medium sustainable (see Table 2).

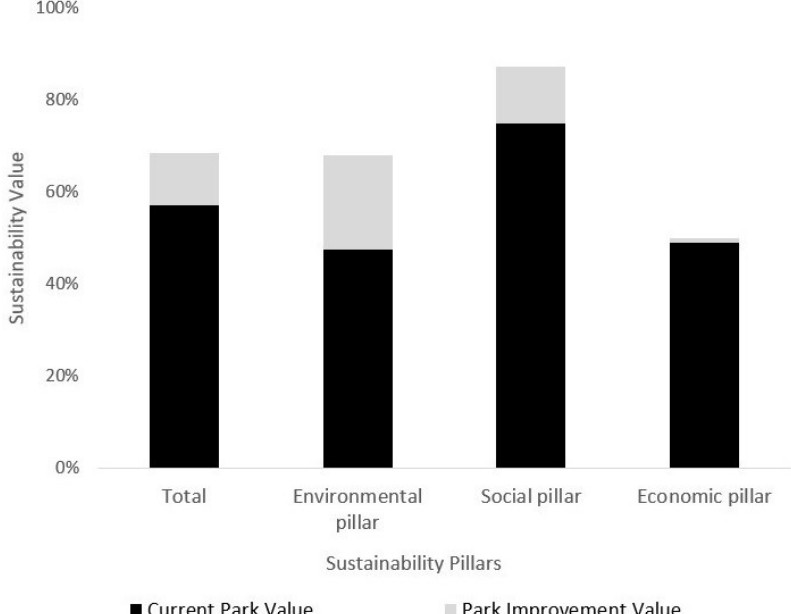

**Figure 6.** Total sustainability value and individual sustainability values obtained per pillar for Cárcamos Park, Mexico.

Table 3 demonstrates the sustainability values of the criteria and indicators for the environmental pillar with 47.7%, Table 4 for the social pillar with 75%, and Table 5 for the economic pillar with 49% sustainability weight. Tables 3–5 also show the respective impact values and the current sustainable park value of Cárcamos Park. The *current park value* is generated from the average of the values of the respective criteria of each indicator.

The park's improvement value is determined by considering internal and external impacts. The park management only influences the internal impacts (i), which is why these are included in the *parks improvement value*. The park management does not directly influence the external impacts (e), which is why the values from criteria with external impacts are not included in the *parks improvement value*. In order to enhance this value, external sources from the park administration, like the municipal or federal government, have to take action.

In the environmental pillar (see Table 3), the indicator with the lowest *current park value* is *sustainable building with green building certification* with 0%. The indicator with the highest *current park value* is *renewable energy and energy efficiency* with 96.5%. None of the indicators reached the highest reachable value of 100. In the same Table 3, the *parks improvement value* of the indicator *(sustainability)* is zero (0) since it is of external impact, while the indicators *renewable energy and energy efficiency* and *waste management* both have a parks improvement value of 100; both are of internal impact.

In the social pillar (see Table 4), the lowest indicators are *environmental policies for the use of green area* with 0% and *(health)* also with 0%. The other indicators all had a current park value of 100%: *exclusive maintenance staff*, *environmental impact on society*, *space for environmental education*, *environmental education workshop*, *environmental management system in office*, and *accessible entrance* .

In the economic pillar (see Table 5), the lowest indicators are *charging fees* and *(efficiency)*, both with 0%. The highest valued indicators are *waste registration and collector control* with 100% and *energy efficiency* with 98%.

A low sustainable value indicates a high area of opportunity. Considering that the pillar with the lowest sustainability level is the environmental pillar with 47.7%, we have identified the indicators with the most significant potential for improvement as follows: E1) *sustainable transport*, (E2) *green area and biodiversity* , (E3) *water conservation*, and (E6) *(sustainability)*.

In second place, the economic pillar obtained a sustainability value of 49%. The indicators *charging fees* (EC2) and *(efficiency)* (EC5) both have the lowest current park value of 0% since the Cárcamos Park charges no fees (entrance, workshop, or rent-space fees) and sustainable construction criteria are not applied. The other areas of opportunity are economic resources generated from the sale of waste (EC1). The park currently collects, separates, and sells 11 types of recyclable materials without considering resale value.

The social pillar (see Table 4) with 75% was the pillar of highest sustainability. Here, the most significant areas of opportunity are *environmental policies for the use of green area* (S5) and *(health)* (S8), both with 0% current park value, since Cárcamos Park has no environmental policy in favor of green spaces. The buildings have no green building certification.

The total sustainability value and the sustainability value obtained for each pillar for Cárcamos are shown in Figure 6. The total sustainability score is determined by summing the points earned by each pillar. In this case study, the park received an overall value of sustainability of 57.2%, indicating a medium level of sustainability with a potential for improvement up to 68.5%. The 31.5% that is missing for 100% sustainability is due to the external factors (e+ or e−) that the park administration cannot change. Only internal impacts (i+ or i−) are factored in the parks improvement value since the park management can directly influence those impacts. Our results appraise a significant opportunity to enhance the sustainability of Cárcamos Park and provide a reference for park managers of Cárcamos parks, helping them in decision-making, prioritizing action implementation, and even justifying requests for economic resources (see Figure 6).

After presenting the results of our methodology's application at Cárcamos Park, the forthcoming section will entail a comparative evaluation of our methodological framework in relation to alternative approaches. This examination will underscore its distinctive strengths while also addressing any potential weaknesses.

## 4. Discussion

In this section, we compare our methodological framework to alternative approaches and discuss its inherent advantages and potential limitations.

Many studies evaluate the perception and satisfaction of urban parks and the experiences and emotions produced within these green spaces, considering the size of the park, the vegetation, the convenient infrastructure, the perception of natural scenery, the conservation of equipment and nature, and the cleanliness of the environment [10,50,51]. While these studies focus mainly on visitor attraction and the appreciation of nature, it is equally essential to consider the parks' contribution to sustainability. Other authors have addressed the issue, attempting to establish the criteria necessary to strike a balance between the essential inputs to the operation of city parks and their benefits: Ávila and Medina [36], for example, analyze different perspectives based on the social-environmental aspects to develop sustainability without including the economic factors. Morales-Cerdas et al. [35] applied 11 environmental indicators for urban green areas to determine the environmental conditions as a tool for urban management, disregarding the importance of the park's economy.

Guerrero and Culós [52] applied six criteria that grouped ten indicators at two case study parks in Argentina. The requirements are reference indicators (the area covered by vegetation and sustainable human load), holistic indicators (the ecological function and heritage index), cause and effect indicators (the depredation of the urban park), projecting indicators (the tourist demand and projected municipal investment in parks), risk and uncertainty indicators (the natural vulnerability and heritage vulnerability), and control and management indicators (the integrated management of the park). Vélez Restrepo [23] shows a conceptual and analytical approach to the sustainability of urban parks and green areas and proposes the construction of a sustainability index based on three principles: (1) the ecological functionality with one indicator, (2) the economy and environmental management of resources with five indicators, and (3) social functionality with three indicators. The main difference compared to our research is that Vélez Restrepo [23] only uses nine indicators, which, in our view, are too abstract and superficial, making it almost impossible for park managers to use them to determine the park's sustainability index and to identify its areas of opportunity.

Instead, our methodology takes into account three pillars: the environmental pillar, the social pillar, and the economic pillar, with each being given equal importance as all three pillars play an essential role in the management of parks. City parks usually depend on the governmental budget, which can be limited. Undoubtedly, economic self-sufficiency can be achieved, for example, through the application of circular economy strategies: Stahel [53] or Geisendorf and Pietrulla [54]. We have included these strategies in our methodology, and park managers can use them as a guide to becoming more sustainable from an environmental, social, and financial perspective. Progressive urban park management must consider and maintain a balance between these three global pillars. This balance was not considered in the reviewed papers and reports, embedded within an easy-to-apply proposal that guides through the necessary operational data represented by our operational criteria (and *additional data*).

Our proposal includes 19 indicators, 50 criteria, and 18 additional data representing easy-to-collect data from the park management (see Table 1). We include data that other researchers do not consider:

1.  The means of transport for employees must be taken into account. If employees come to the park by car, this harms the sustainability index. If they arrive instead on foot, by bike, or by public transport, the impact on sustainability would be positive.
2.  We include waste management. It is vital that a sustainable park offers visitors a waste separation infrastructure and a waste collection center. Organic waste may end up in the compost as fertilizer for the park's greenery areas. Other waste like metal or PET can be sold and help to improve the park's economy.
3.  A sustainable park needs environmental policies that give instructions for using green areas. Policies for saving water and energy are necessary and encourage

the sustainable behavior of park visitors and employees and are beneficial for the sustainable development of the park ([55,56]).

4.　Another critical area of sustainability is energy efficiency. Here, we must consider using (a) renewable energies such as sun, wind, or biomass; and (b) illumination in the park, green spaces, and offices. The energy consumption caused by the illumination is an essential issue for a sustainable approach [57].

5.　Our methodology includes the level of biodiversity as an indicator to define the sustainability of a park. The reason is that even small green spaces such as parks can include biodiversity if they provide water bodies (ponds or lakes) and green spaces nearby, creating a natural green space network [58].

6.　We include the proportions of native and exotic species and their health conditions since local healthy species have to be favored [42]. Identifying tree species is a relatively easy task that an observant park employee can accomplish as the trees remain visibly in place.

7.　Our methodology also integrates the maintenance workforce and the environmental impact on society through environmental education, and city parks can help reduce crime in their sphere of influence [59].

8.　Our proposal includes information on the space occupied by buildings or parking facilities, which have a negative impact since they reduce the permeable area, increase waste generation, and raise energy consumption [60].

9.　Our proposal includes the sustainability status of buildings inside the park. Buildings with green building certification or rating tools lead to environmental, economic, and social benefits due to their sustainable construction materials and energy efficiency [44,48].

10.　Our methodology considers the economic aspects separately from the social and environmental aspects, seeking economic self-sufficiency from the services that the park can provide to the citizens, not only by saving money efficiently but also by generating money from selling recyclable waste collected and separated at a collection center inside the park, offering workshops and renting space inside the park.

Sturiale and Scuderi [61] merge the economic and social aspects to "eco-social". They consider the dimensions of sustainable development to contribute to promoting a governance model for the city called "eco-social-green". Indeed, the economic and the social aspects are strongly related in some ways. For example, Cárcamos Park does not generate any economic resources from the visitors since no fees are charged inside the park. For this reason, the current park value generation of financial help from the *charging fees* (EC2) is zero. It is essential to mention that, for example, a low entrance fee means, on the one hand, a low economic value but, on the other hand, a high social value since access to the park is facilitated to everyone, regardless of their financial income and therefore favoring everyone's well-being. The same applies to fees charged for workshops or space the park offers.

It must also be mentioned that the size of a park and its density of tree cover positively impact visitors' perception and promote more visits [50]. Larger urban parks receive more visitors than smaller parks, and the size is more important than the distance a park visitor has to travel to go to the park. The fact that larger parks attract more visitors, regardless of distance, could be detrimental to sustainability, given emissions from traffic. Therefore, a minimum green surface area of the park and the surrounding infrastructure, including the park's connection to public transport and bicycle lines, should be regarded [62].

In 2020, Cárcamos Park received an average of 171 visitors per day, considering only the seven months after its opening because of the COVID-19 pandemic. Our methodology does not include the number of visitors per day since the number itself is unimportant. More important is how those visitors get to the park without contaminating the services (recycling center or educational programs) they use, and what they learn from their visit and their behavior inside the park.

Urban parks help to conserve the local biodiversity and can be home to wildlife [63]. Besides the simple distinction between native and exotic plant species, our methodology does not include any distinction of wild animals such as birds, rodents, or amphibians,

which could be of interest for the sustainability of urban parks, as mentioned in [64]. The reason for this is that our methodology is applicable to any park worker. This distinction would involve detailed biological research, elaborated on by external experts, since animals move or hide and may be difficult to find or identify, especially insects.

Dizdaroglu [13] considers healthy food as an indicator for sustainable parks. At this stage, our methodology does not include offering, consuming, or promoting healthy food since our focus includes general sustainable data leading to the parks' resilience. However, this aspect might be considered and included in the future.

Dizdaroglu [65] and Dizdaroglu [13] mention the importance of good governance, which includes the consolidation of democratic institutions at all levels to ensure transparency and accountability in governance and inclusive participation in decision-making.

During the case study analysis, potential avenues were identified to be developed and included in the database. In the future, our methodology can be expanded to include additional criteria:

- More transport indicators, i.e., how visitors arrive at the park,
- The human carrying capacity, i.e., how many visitors the park can support to remain balanced.
- Governance indicators, i.e., assess the transparency and accountability in governance and inclusive participation in decision-making.

This article presents a new methodology that is easy to apply by any park manager to parks of any size. The goal is to assign the sustainability grade and show new ecologic, social, and economic areas of park opportunities. This methodology is balanced as it gives equal weight to the three pillars of sustainability: environmental, economic, and social. Its application can identify the strengths and weaknesses of a park's sustainability. The method emerges from a detailed analysis of the park's operations. It incorporates criteria and indicators that have allowed us to assign measurable values and measure sustainability quantitatively. This will facilitate park administrators making the right decisions in the future.

## 5. Conclusions

In conclusion, our presented methodology represents a valuable and accessible tool for enhancing the sustainability of urban parks. It offers a simple and cost-effective approach to identify improvement opportunities, establish baseline parameters, and guide the allocation of economic resources while justifying their augmentation.

This methodology's significance lies in its ability to bridge the gap between technical complexity and practical application. It employs plain language and basic technical parameters, making it accessible to park managers without requiring expensive consulting services. Users of this methodology benefit from a comprehensive view of park operations, with 19 indicators generated from 50 operational criteria and 18 additional data points. This holistic perspective empowers park managers to assess operational efficiency, make informed decisions, and benchmark their park's performance against others, thereby facilitating the replication of successful practices.

The successful demonstration of this methodology at Cárcamos Park in León, Mexico, resulting in a sustainability score of 57.2%, underscores its practicality. This score falls within the range of average sustainability, between 51% and 79%, and demonstrates its relevance in assessing and enhancing park sustainability. Scores below 51% indicate low or negligible sustainability, while those exceeding 79% signify a high level of sustainability.

In summary, our proposed method represents a dependable tool for evaluating the sustainability status of urban parks. Beyond its immediate applicability, it lays the foundation for further research in sustainable building certification and green area management, including $CO_2$ capture assessment. Looking forward, we envision automating the database to streamline the assessment process, enabling users to upload information effortlessly and receive automatic results like diagrams and tables for straightforward cross-park comparisons. Additionally, this methodology can serve as a reference for establishing a ranking

system for urban parks, fostering collaboration among park managers to refine and enhance its effectiveness continually.

**Author Contributions:** Conceptualization, T.G. and P.B.; methodology, T.G.; validation, P.B., C.N.S. and F.M.; formal analysis, T.G., P.B., F.M. and C.N.S.; investigation, T.G.; resources, T.G.; data curation, T.G.; writing—original draft preparation, T.G., P.B. and C.N.S.; writing—review and editing, T.G., P.B., F.M. and C.N.S.; visualization, C.N.S.; supervision, P.B.; project administration, P.B.; and funding acquisition, P.B. All authors have read and agreed to the published version of the manuscript.

**Funding:** The APC was funded by Universidad Panamericana campus Aguascalientes.

**Institutional Review Board Statement:** The study was conducted according to the guidelines of the Declaration of Helsinki and approved by the Ethics Committee of Cárcamos Park (protocol code DGMA-CE-2023-GTO-002 approved on 30 August 2023).

**Informed Consent Statement:** Informed consent was obtained from all subjects involved in the study.

**Data Availability Statement:** Not applicable.

**Acknowledgments:** The authors thank Cárcamos Park managers for providing us with the data to perform a sustainability analysis of the park.

**Conflicts of Interest:** The authors declare no conflict of interest.

## Abbreviations

The following abbreviations are used in this manuscript:

| | |
|---|---|
| E1–E6 | Indicators of the environmental pillar |
| S1–S8 | Indicators of the social pillar |
| EC1–EC5 | Indicators of the economic pillar |
| e+ | External dependency with positive impact |
| e− | External dependency with negative impact |
| i+ | Internal dependency with positive impact |
| i− | Internal dependency with negative impact |
| LED | Light-emitting diode |
| PET | Polyethylene terephthalate |
| MDPI | Multidisciplinary Digital Publishing Institute |
| INEGI | National Institute of Statistic and Geography |

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
