# Peer review of "Sustainability of Urban Parks: Applicable Methodological Framework for a Simple Assessment"

_sustainability, doi:10.3390/su152115262_

Round 1
Reviewer 1 Report
Define the parks to which this study applies.
First of all, how large is the park to apply these indicators? Green spaces are limited in large cities and densely populated areas. For example, there are many small parks of 100-200m2.
In addition, the importance of indicators varies depending on the purpose and characteristics, such as a park that specializes in grounds, a park that is mostly grass, or a park that has a sidewalk around a pond.
Next, I have a question about the definition of indicators.
Indicator: 625 hectares of trees is meaningless, affected by tree height and diameter.
Is it the ecological basis of the park that 80% of native species are good?
How many healthy trees? The health of trees is managed in parks.
Minor editing of English language required.
Reviewer 2 Report
see the attachment

Moderate editing of English language required
Reviewer 3 Report
You wrote a very interesting article and you created a good study and analysis tool for urban park managers.
Although it is not a closed model, as, in the future, other researchers will adapt it having in mind the development of society and of the planet Earth, as a starting point it is a very good model considering the defined pillars, indicators and criteria.
Small remarks
p. 6, 226 - not criteria but criterion
p.11, 467 - not 68.5.0% but 68.5%
Round 2
Reviewer 1 Report
Cited documents for park definitions are reports, not papers. As legal interpretations differ in each country, the definition used in this paper is fine, but it cannot be considered to apply to all parks. There is little evidence that it can be adapted to small parks, lawns, sports parks, waterside parks, etc. If it cannot be shown with evidence, this paper will be limited to large parks.
No answer to comment 3.
In the case of a forest landscape, a DBH of 40 cm, a tree height of 20 m, and 200 trees per ha will have a high visual effect.
There is no evidence for 625trees/hr of national regulations.
Reviewer 2 Report
incorporated all the comments
Minor editing of English language required
Round 3
Reviewer 1 Report
Accept in present form.